# Pulp–Dentin Complex Regeneration with Cell Transplantation Technique Using Stem Cells Derived from Human Deciduous Teeth: Histological and Immunohistochemical Study in Immunosuppressed Rats

**DOI:** 10.3390/bioengineering10050610

**Published:** 2023-05-19

**Authors:** Larissa Regina Kuntze dos Santos, André Antonio Pelegrine, Carlos Eduardo da Silveira Bueno, José Ricardo Muniz Ferreira, Antonio Carlos Aloise, Carolina Pessoa Stringheta, Elizabeth Ferreira Martinez, Rina Andréa Pelegrine

**Affiliations:** 1Faculdade São Leopoldo Mandic, Instituto de Pesquisas São Leopoldo Mandic, Endodontia, Campinas 13045-755, Brazil; 2Faculdade São Leopoldo Mandic, Instituto de Pesquisas São Leopoldo Mandic, Implantodontia, Campinas 13045-755, Brazil; 3RCrio Bioengenharia, Campinas 13098-324, Brazil; 4Faculdade São Leopoldo Mandic, Instituto de Pesquisas São Leopoldo Mandic, Patologia Oral, Campinas 13045-755, Brazil

**Keywords:** regeneration, dental pulp, transplants, stem cells, immunohistochemistry

## Abstract

The aim of this study was to histologically verify the performance of pulp-derived stem cells used in the pulp–dentin complex regeneration. Maxillary molars of 12 immunosuppressed rats were divided into two groups: the SC (stem cells) group, and the PBS (just standard phosphate-buffered saline) group. After pulpectomy and canal preparation, the teeth received the designated materials, and the cavities were sealed. After 12 weeks, the animals were euthanized, and the specimens underwent histological processing and qualitative evaluation of intracanal connective tissue, odontoblast-like cells, intracanal mineralized tissue, and periapical inflammatory infiltrate. Immunohistochemical evaluation was performed to detect dentin matrix protein 1 (DMP1). In the PBS group, an amorphous substance and remnants of mineralized tissue were observed throughout the canal, and abundant inflammatory cells were observed in the periapical region. In the SC group, an amorphous substance and remnants of mineralized tissue were observed throughout the canal; odontoblasts-like cells immunopositive for DMP1 and mineral plug were observed in the apical region of the canal; and a mild inflammatory infiltrate, intense vascularization, and neoformation of organized connective tissue were observed in the periapical region. In conclusion, the transplantation of human pulp stem cells promoted partial pulp tissue neoformation in adult rat molars.

## 1. Introduction

Dental pulp inflammation or infections are commonly treated by removing the pulp tissue and replacing it with inert materials or bioactive materials (e.g., gutta-percha and sealer cement) via endodontic treatment. Pulp removal makes endodontically treated teeth become devitalized. An alternative approach to standard endodontic treatment is the use of tissue engineering concepts. The evolution of research in the field of tissue engineering has shown that the triad composed of stem cells, growth factors, and scaffold is the key to success [1,2]. Since the discovery of stem cells in the second half of the 20th century, which are cells that may remain undifferentiated for an extended period of time and can renew and differentiate as necessary, there have been significant advancements in the field. Several studies have shown the possibility for stem cells to regenerate different tissues, such as skin [3,4], muscle [5,6], fat [7,8], cartilage [9,10], and bone [11,12], among others. Moreover, despite major challenges, the reconstruction of organs such as the trachea [13], esophagus [14], kidney [15], and teeth [16] was developed by using tissue engineering concepts and cell therapy. 

For cell therapy and tissue engineering purposes, there are many possibilities concerning the source tissue, and its selection should consider the characteristics of the tissue to be reconstructed. In this regard, Coelho de Faria et al. [17] showed that mesenchymal stem cells derived from bone marrow were more prone to induce osteogenic differentiation than mesenchymal stem cells derived from adipose tissue, despite the fact that both source tissues have the same mesenchymal origin. In this sense, there is a tendency in the scientific literature to choose the pulp as the source tissue for pulp–dentin complex regeneration.

Endodontics has been bringing new proposals for techniques and materials, increasing its effectiveness and reducing operative risks. Despite the high success rates in the preservation of the dental element, the conventional endodontic treatment causes permanent devitalization of the treated tooth, rendering it insensitive and more susceptible to fracture [18,19]. Regeneration of the pulp–dentin complex is an alternative to conventional treatment and consists of reconstituting damaged tissue by forming histological structures similar to original structures, and restoring its sensorial, nutritive, formative, reparative, and defensive functionality [20,21]. For pulp–dentin complex regeneration, it seems reasonable to select the pulp or apical papilla as a source tissue, as idealized by Kim et al. [22] and Sequeira et al. [23], respectively.

Pulp regeneration has been investigated through two models: (1) cell homing or (2) stem-cell transplantation. In the first model, a scaffold containing chemotactic signaling molecules is packaged in a previously prepared root canal so that progenitor cells are recruited into the canal [24,25]. In the second model, stem cells are added to a scaffold, which is then directly transplanted into a previously prepared dental root canal [26,27,28].

Research with orthotopic models has been successfully conducted. In one such study, the tissue of pulpectomized dog teeth was completely regenerated after receiving autologous pulp stem cells, mobilized by adding a granulocyte colony-stimulating factor [27]. The methodology of this study served as the basis for the first pilot study of autologous dental pulp stem cell transplantation in humans, performed with five patients with irreversible pulp inflammation (i.e., pulpitis). In most of these patients, a positive response to electrical pulp tests could be observed (i.e., vital pulp tissue). Moreover, resonance imaging scans showed similar results between normal pulp and engineered pulp–dentin complex, such as dentin formation [28]. However, more research using animal models is needed to ensure that pulp regeneration induced by stem-cell transplantation is safe for application in dental practice, especially in mature teeth [22,28]. Thus, the aim of the present study was to perform a histological and immunohistochemical evaluation of the regeneration of the pulp–dentin complex promoted by transplanting stem cells from the dental pulp of human deciduous teeth into molars of adult rats with fully formed apices. The null hypothesis of this research was a lack of improvement in the pulp–dentin complex regeneration when pulp-derived stem cells were used.

## 2. Materials and Methods

Cryopreserved pulp stem cells from human deciduous teeth were donated by the R-Crio laboratory (Campinas, SP, Brazil), pursuant to an approval issued by the National Research Ethics Commission (CAAE: 37183020.8.0000.5374), National Health Council. The sample size was determined according to a previous study conducted by Nabeshima et al. [24] and calculated based on a type I error (α) of 5%, a type II error (β) of 20%, a power of 80%, and an effect size of 1.19, thus rendering a minimum sample size of 12 specimens.

### 2.1. Selection and Preparation of Animals

Twelve adult male Rattus norvegicus albinus rats (Wistar strain), aged 10 weeks and weighing approximately 300 g, were selected following approval of the study by the Animal Use Ethics Committee of the institution where the study was conducted (protocol no. 2020/22). All the experiments complied with ARRIVE (Animal Research: Reporting of In Vivo Experiments) guidelines and were carried out in accordance with the National Research Council’s Guide for the Care and Use of Laboratory Animals. Considering the possibility of human cell rejection, an immunosuppression protocol was started 10 days before the experiment, as described by Lekhooa et al. [29], and followed until euthanasia. The animals were intraperitoneally anesthetized with a 10% ketamine hydrochloride solution (Quetamina; Vetnil, Louveira, SP, Brazil; 70 mg/kg), combined with a 2% xylazine hydrochloride solution (Sedanil; Vetnil; 10 mg/kg). The animals were maintained in the supine position on a wooden platform, and orthodontic elastics were attached to the incisor teeth of both arches on one side, and to hooks on the surface of the platform on the other side to keep their mouths open. Small retractors made with conveniently bent orthodontic wire no. 08 (Morelli, Sorocaba, SP, Brazil) were used to retract their cheeks.

The 24 maxillary first molars of the 12 animals were divided into two groups (n = 12): the SC group, in which the mesiobuccal canals of the teeth on the left side received transplanted stem cells delivered in a phosphate-buffered saline solution (PBS); and the PBS group, in which the MB canals of the homologous teeth on the right side were filled only with PBS.

### 2.2. Endodontic Procedures

The presence of fully formed roots was radiographically confirmed. The pulp chamber was accessed with a 1/2 spherical carbide bur (Microdont, São Paulo, SP, Brazil) at high speed and filled with a 1% sodium hypochlorite (NaOCl) solution (Asfer, São Caetano do Sul, SP, Brazil). The mesiobuccal canal was located and explored with a #10 K-type file (Dentsply Maillefer, Ballaigues, Switzerland). The working length was determined using a foraminal locator (Root ZX II; J. Morita, Kyoto, Japan) and set 1 mm short of the apical foramen. The root canal was then instrumented with a primary file (25/0.07) of the WaveOne Gold system (Dentsply Maillefer), driven by an endodontic motor (X-Smart Plus; Dentsply Maillefer) in a reciprocating motion. Three in-and-out movements were used at the cervical, middle, and apical levels until the working length was reached. The canal was flushed prior to each movement with 2 mL of 1% NaOCl using a 30-gauge NaviTip needle (Ultradent, South Jordan, UT, USA) attached to a 3-mL hypodermic syringe, positioned 2 mm short of the working length, and foraminal patency was maintained with a #10 K-type file (Dentsply Maillefer). After instrumentation, the canal was irrigated with 2 mL of a 17% ethylenediaminetetraacetic acid (EDTA) solution (Formula e Ação, São Paulo, SP, Brazil) for 1 min, and finally with 2 mL of 1% NaOCl. Final aspiration was performed with a capillary tip (Ultradent), and the canal was dried with absorbent paper points (Dentsply Maileffer).

### 2.3. Stem-Cell Transplantation

A deciduous tooth was extracted and taken to the laboratory. All laboratory procedures were performed at R-CrioCriogenia, Campinas, Brazil, in a laboratory classified as ISO7 (ISO 14644). At the lab, the pulp was collected, and the sample was rinsed in a solution containing 100 U/mL penicillin/streptomycin (Sigma, St Louis, MO, USA), followed by enzymatic digestion with 1 mg/mL collagenase type I at 37 °C for 5 min. The digestion was stopped by the addition of a low-glucose DMEM basal medium (Sigma, USA) and centrifuged at 178× *g* for 10 min. The supernatant was discarded, and the pellet was washed with 1× PBS buffer to remove reaction residue and centrifuged again at 178× *g* for 10 min. The supernatant was discarded, and the pellet was suspended with DMEM supplemented with 10% (*v*/*v*) fetal bovine serum (Sigma, USA—Cat. F2561), 1% (*v*/*v*) Lglutamine (Sigma, USA: Cat 59202C), and 1.1% (*v*/*v*) penicillin/streptomycin (Sigma, USA: Cat P4333). This content was inoculated into a 25 cm^3^ bottle, and the cells were incubated at 37 °C and 5% CO_2_ (Panasonic, MCO-19AIC UV). The culture medium was replaced with a new aliquot every 72 h. Upon reaching 65 to 75% confluence, the cells were enzymatically retrieved (trypsin) for cell passage.

The pulp stem cells obtained by the protocol cited above were supplied in a PBS suspension (1 × 10^6^ cells per mL). In the SC group, approximately 25 µL of the suspension was inserted into the prepared canal using a 30-gauge NaviTip needle (Ultradent) attached to a 1-mL syringe. A Teflon barrier (Isotape; TDV, Pomerode, SC, Brazil) was then placed over the suspension, and the cavity was sealed with a layer of cement based on mineral trioxide aggregate (MTA; Angelus, Londrina, PR, Brazil) and a subsequent layer of light-cured glass ionomer (Ionofast; Biodina, PR, Ibiporã, Brazil). In the PBS group, the procedures were the same, but the canal was filled only with PBS. After 12 weeks, the animals were intraperitoneally euthanized by saturation with 2% isoflurane (Isoforine; Cristália, São Paulo, SP, Brazil).

### 2.4. Histological and Immunohistochemical Processing

Block sections of the maxillae were removed and fixed in 10% buffered formalin for 24 h, decalcified in a 20% formic acid solution for 10 days, and washed under running water for 8 h. They were then dehydrated in graded ethanol concentrations, cleared in xylol, and embedded in histologic paraffin. Histological evaluation was performed by staining serial 4-μm longitudinal sections with hematoxylin-eosin and then mounting them on glass slides using biological resin (Permount Mounting Medium; Fisher Scientific, Fair Lawn, NJ, USA). 

Immunohistochemical evaluation was performed by deparaffinizing, hydrating, and immersing 3-µm longitudinal sections in a 3% hydrogen peroxide solution for 30 min (Dinâmica, Diadema, SP, Brazil). Antigen retrieval was performed in a sodium citrate buffer solution (pH 6.0) for 1 h (Sigma, St Louis, MO, USA). Subsequently, the sections were incubated overnight at 4 °C with the primary antibody to dentin matrix protein 1 (rabbit polyclonal anti-DMP1, 1:75; Takara Bio, Kusatsu, Shiga, Japan), and subsequently with the secondary antibody (EnVision System; Dako, Carpinteria, CA, USA). The sections were stained for 5 min at 37 °C with 3,3′ diaminobenzidine tetrahydrochloride (DAB; Dako), and counterstained with hematoxylin (Dinâmica). Slide images were captured using a computerized acquisition system (AxioVision Rel. 4.8; Carl Zeiss, Oberkochen, Germany) coupled to a light microscope (Axioskop 2 Plus; Carl Zeiss), and evaluated by a single examiner, blinded to the group assignment of the specimens. 

The histological analysis involved the observation of intracanal connective tissue, odontoblast-like cells, intracanal mineralized tissue, and periapical inflammatory cell infiltrate. The immunohistochemical analysis ascertained whether or not there was cytoplasmic immunostaining for DMP1.

## 3. Results

In the PBS group, an amorphous substance and remnants of mineralized tissue were observed, both distributed throughout the root canal. Abundant inflammatory cells, predominantly of the polymorphonuclear type, and a small number of blood vessels were observed in the periapical region (Figure 1). 

In the SC group, an amorphous substance and remnants of mineralized tissue were observed, both distributed throughout the root canal. However, odontoblast-like cells were observed in the apical region of the canal in addition to material with a basophilic aspect compatible with the formation of a mineral plug. In the periapical region, inflammatory cells were observed, albeit in smaller numbers than in the PBS group, characterizing a mild polymorphonuclear inflammatory infiltrate, with numerous blood vessels and newly formed organized connective tissue (Figure 2).

Immunohistochemical analysis revealed the presence of DMP1-positive odontoblast-like cells in the apical region of the canal in specimens from the SC group (Figure 3A–C) and DMP1-negative cells in the PBS group (Figure 3D–F).

## 4. Discussion

The present study used human pulp stem cells from deciduous teeth because they are easy to obtain, have a high capacity for proliferation, high plasticity, and a high pro-angiogenic effect [30,31]. This technique was also preferred because pulp cells introduced into the root canal render the neoformation of innervated and vascularized pulp tissue more predictable, potentially supplying active odontoblasts capable of producing dentin [32]. Moreover, for pulp–dentin complex regeneration, it seems reasonable to select the pulp as a source tissue as it matches the tissue to be reconstructed. This concept was used by other studies [22]. 

Despite the fact many authors have used simpler methods of cell concentration for tissue engineering purposes [33,34,35,36] (e.g., centrifugation with or without a density gradient), the cell culture method was selected in the present study. As pulp tissue volume is relatively small, it is hard to obtain an adequate number of stem cells without extensive culture passages. On the other hand, tissues such as bone marrow and fat can be obtained in huge amounts, allowing the use of simpler methods such as the bone marrow aspirate concentrate (or the bone marrow mononuclear fraction) and stromal vascular fraction of adipose tissue, respectively. Therefore, the use of the isolation, proliferation, and cultivation of mesenchymal stem cells methodology seems to be a trend in the scientific literature in situations where the pulp is selected as the source tissue for tissue engineering purposes. 

Research in animal models is essential to test the safety and efficacy of stem-cell transplantation [37]. In addition, it allows for performing accurate histological and immunohistochemical analyses of the components of the newly formed tissue. In contrast, assessments based on human research using sensitivity tests, periapical radiographs, and imaging tests [28] have no way of positively identifying the resulting tissue composition. In this regard, the use of rats seems to be adequate as it is a well-documented experimental model for both histological and immunohistochemical analyses. Ethical release, fast reproducibility, low cost, high similarity to the human genome, and speed in obtaining results seem to be ideal conditions for choosing the animal for research. Thus, the animal chosen as a model in this research was the rat, as it meets most of these requirements. However, due to the small size of rats’ oral cavity, in the present study, huge attention was paid concerning the positioning and anesthesia protocol. Moreover, the use of visual magnification, using delicate instruments (e.g., small burs), and lip, tongue, and cheek retractors seemed to be mandatory. 

The orthotopic model used herein and in previous studies [22,37] has proven capable of producing more reliable assessments than both ectopic models (e.g., stem cells implanted into the subcutaneous region or in the renal capsule of rats) [38,39] and semi-orthotopic models (e.g., tooth slices or fragments) [40,41]. However, the importance of ectotopic and semi-orthotopic models for scientific knowledge cannot be overlooked. In this regard, Yu et al. [39] proved, in an ectopic model in rats, that mesenchymal stem cells derived from pulp tissue have greater odontogenic capacity than bone marrow mesenchymal stem cells. In their study, the mesenchymal stem cells, after being isolated from the dental pulp and bone marrow of rats, were cultured in an induction microenvironment produced by cells from the apical papilla of incisors of two-day-old rats. The analysis showed that pulp tissue mesenchymal stem cells were able to generate typical dental tissues, showing amelogenesis and dentinogenesis, while bone marrow mesenchymal stem cells produced only an osteodentin structure without enamel formation. Therefore, once more, the selection of the source tissue was shown to be of major importance for tissue engineering purposes. This concept reinforced the selection of the pulp as the source tissue to achieve pulp–dentin complex regeneration in the present study.

Swift results and ethical approvals, reproducibility, and low cost are the ideal conditions for using research animals [22,42]. The rat was chosen in the present study because it met all of these requirements, and the intervention was performed after 10 weeks of life, considering that this is the average time required for the complete formation of the root apex in these animals [43]. The mesiobuccal root of the maxillary first molar was chosen because of its great buccal inclination, which facilitated access to its canal. Furthermore, its canal is easily distinguishable from the other canals of this tooth, and the diameter of this canal is similar to that of human teeth [24], thereby allowing the use of files commonly used in clinical practice. This decision makes the translational of the current results to the clinical practice easier, which is also, whenever possible, a demand in animal studies. 

In the present study, the unique factor that does not match with current endodontic treatment was the filling materials used (i.e., PBS and stem cells). All other procedures adopted were based on the current standard knowledge about clinical endodontics. The NaOCl solution concentration chosen for irrigation during instrumentation was 1%, given its tolerable cytotoxicity [44,45], which may not significantly impair the transplanted stem cells’ activity. A NaOCl solution concentration of 6% significantly decreased the survival and expression of dental pulp stem cells in an in vitro study [45]. In the present study, besides the 1% NaOCl solution, a 17% EDTA solution was also used after instrumentation, given its potential to promote stem-cell adhesion, migration, and differentiation [46,47]. Both the 1% NaOCl and 17% EDTA solutions are currently used in daily practice. 

The selection of the source tissue used in the present study was based on the characteristic of the tissue that was intended to be reconstructed (i.e., pulp tissue). Despite the fact that mesenchymal stem cells from different sources can theoretically generate all mesenchymal tissues, previous findings showed that the regenerative results can be different among different sources and target tissues, suggesting a desirable match between the source tissue for cell therapy and the tissue to be reconstructed. In this sense, Coelho de Faria et al. [17] stated that mesenchymal stem cells derived from bone marrow are more prone to differentiate into the osteoblastic lineage than mesenchymal stem cells derived from adipose tissue. On the other hand, Paul et al. [48] verified that, in myocardial infarction situations, sites where adipose-derived stem cells were used showed higher potential in adopting cardiomyocyte phenotype when compared with bone marrow-derived stem cells. In this regard, Shaikh et al. [49] showed that the use of human umbilical cord mesenchymal stem cells was effective in periodontal regeneration. However, from a translational point of view, this source tissue does not seem to be a candidate for autogenous use, as it demands the cryopreservation of stem cells for a long period of time. 

However, it is important to state that, in the present study, xenogenous cells were used instead of the gold-standard autologous cells. Nonetheless, the lack of autologous cell usage was minimized by the selection of immunosuppressed animals, such as in another study in the tissue engineering field [48]. Therefore, in this perspective, the bias concerning the use of xenogenous cells was significantly attenuated and made it possible to test human cells, which are more adequate from a translational point of view. 

Histological analysis revealed the presence of an amorphous substance and mineralized tissue in both groups. The amorphous substance likely represents pulpal remnants from root canal walls untouched by the files during instrumentation. This finding confirms the results of microtomographic studies that found percentages of 11 to 50% of untouched areas after the use of various instrumentation systems [50,51]. Remnants of mineralized tissue compatible with dentin shavings also confirm previous studies showing that dentinal debris accumulation during instrumentation is a common finding [52,53]. Robinson et al. [52], by comparing the dentinal debris levels of a single instrument technique in reciprocating kinematics (Wave One) against the rotary technique using multiple instruments (Pro Taper), verified debris levels of 19.5 and 10.5%, respectively. 

Histological analysis also revealed partial tissue neoformation in the SC group. This was confirmed by the immunohistochemical analysis, which revealed a positive expression of DPM1, an acidic phosphoprotein responsible for the differentiation of pulp stem cells into odontoblasts [54]. Therefore, the null hypothesis of the present study was rejected. However, this result does not represent complete regeneration, as reported by Nakashima and Iohara [27], who observed a regenerated and functional pulp tissue. On the other hand, it is also distinguishable from that reported by Zhu et al. [26], who observed the formation of a periodontal-like tissue, without the presence of odontoblast-like cells. 

In this sense, it is important to state that regenerative therapy is not only the replacement of tissue where there was injury, but also necessary to recover physiological functions, such as the original tissue. Therefore, in the present study, complete pulp–dentin complex regeneration was not observed. However, in a clinical pilot study with five humans [28] aiming to regenerate previously excised pulps, the use of mesenchymal stem cells derived from the pulp of wisdom teeth in the atelocollagen scaffold showed adequate response to electric pulp tests. After twenty-four weeks, the signal intensity on magnetic resonance imaging (MRI) was similar to that of the normal pulp in four patients, and the computed tomography (CT) scan demonstrated dentin formation in three of the five patients. There was only one case of failure, with periodontal ligament thickening on radiography at twelve weeks and periapical radiolucency at twenty-four weeks. Nonetheless, despite the fact of these significant results in this pilot clinical study, it is not possible to claim that real pulp–dentin complex regeneration occurred in the four patients, as there is no possibility to identify the resulting tissue composition in this experimental model. 

In the present study, areas of highly mineralized basophilic tissue lacking dentinal tubule-like structures were observed in the apical region of the root canal. This tissue more closely resembled cementum or bone rather than dentin, and suggested the formation of a mineral plug, a finding frequently observed following the application of regenerative endodontic therapies [55,56,57]. 

In the PBS group, abundant inflammatory cells and few blood vessels were observed in the periapical region, whereas fewer inflammatory cells, intense vascularization, and connective tissue neoformation were observed in the same region in the SC group, characterizing a reparative tissue. This more favorable picture can be attributed to the ability of transplanted pulp stem cells from deciduous teeth to suppress inflammation by inhibiting the secretion of pro-inflammatory cytokines [58]. These findings, taken together with the partial tissue formation observed only in the SC group, can also be attributed to the ability of transplanted stem cells to differentiate in other cell lineages, which can contribute to tissue regeneration.

Adequate blood supply is of great importance to promote tissue regeneration, considering its ability to promote angiogenesis and ensure the nutrition of stem cells in a timely fashion. The small size of the apical foramen can render this supply difficult, especially in molars [58], hence impairing cell nutrition throughout the entire length of the root canal, and potentially explaining the present observation of odontoblasts solely in the apical region. 

The purpose of the present study was to outline a working protocol for an orthotopic model of stem-cell transplantation into the root canals of rat molars. Its limitations include technical difficulties related to the small dimensions of rat teeth and the scarcity of available studies using similar methodologies, for comparison purposes. Future studies are warranted using different scaffolds or vehicles for insertion and maintenance of stem cells in the minute space of the root canals, stem cells from other sources—such as the periosteum, where they are easy to collect even in adults—and, lastly, growth factors capable of contributing to promoting angiogenesis and neurogenesis. For these purposes, more animal studies are certainly required; in addition, future clinical studies would be able to validate the real clinical benefits of cell therapy and tissue engineering approaches.

## 5. Conclusions

In this experimental study, the transplantation of pulp stem cells promoted partial tissue neoformation of the pulp–dentin complex. In addition, mild inflammatory infiltrates, intense vascularization, and organized connective tissue were observed in the periapical region.

## Figures and Tables

**Figure 1 bioengineering-10-00610-f001:**
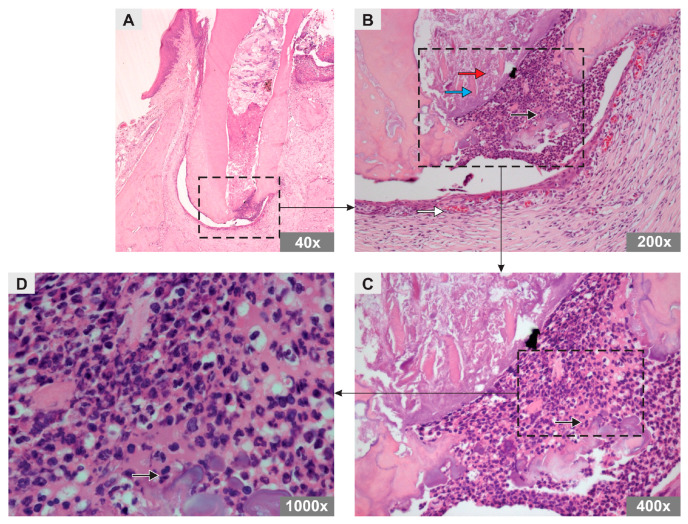
Photomicrograph representative of the longitudinal sections of the mesiobuccal root of maxillary first molars of rats in the PBS group (H&E staining). Original magnification (**A**) 40×, (**B**) 200×, (**C**) 400×, (**D**) 1000×. Note the arrows representing an amorphous substance (red arrow), mineralized tissue remnants (blue arrow), polymorphonuclear inflammatory cells (black arrow), and blood vessels (white arrow).

**Figure 2 bioengineering-10-00610-f002:**
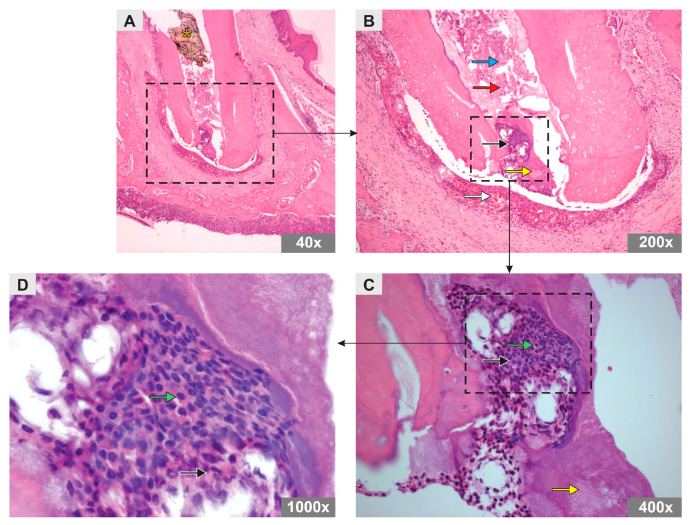
Photomicrograph representative of the longitudinal sections of the mesiobuccal root of maxillary first molars of rats in SC group (H&E staining). Original magnification (**A**) 40×, (**B**) 200×, (**C**) 400×, (**D**) 1000×. Note the arrows representing an amorphous substance (red arrow), mineralized tissue remnants (blue arrow), polymorphonuclear inflammatory cells (black arrow), blood vessels (white arrow), odontoblast-like cells (green arrow), and mineral plug (yellow arrow). Asterisk shows the Teflon barrier.

**Figure 3 bioengineering-10-00610-f003:**
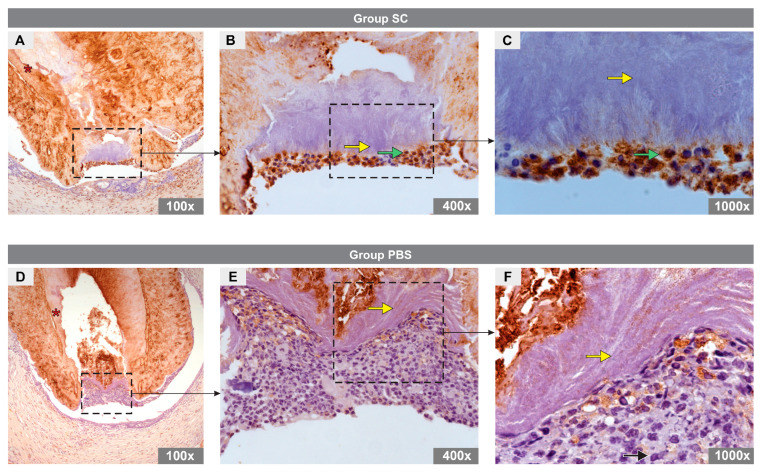
Immunoexpression of DMP1 in the experimental groups. (**A**–**C**) images representative of specimens from SC group and (**D**–**F**) images representative of specimens from PBS group. In the SC group, note the cytoplasm of odontoblast-like cells marked for DMP1 (green arrow), dentin (asterisk), and mineral plug (yellow arrow). No DMP1 expression was observed in the PBS group. Note the detin (asterisk) and mineral plug (yellow arrow).

## Data Availability

Data supporting reported results can be acquired by email.

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
