# Peer review of "Pulp–Dentin Complex Regeneration with Cell Transplantation Technique Using Stem Cells Derived from Human Deciduous Teeth: Histological and Immunohistochemical Study in Immunosuppressed Rats"

_bioengineering, 2023, doi:10.3390/bioengineering10050610_

Round 1
Reviewer 1 Report (Previous Reviewer 1)
Title: Pulp-dentin complex regeneration by cell transplantation technique using stem cells derived of the human deciduous teeth: histological and immunohistochemical study in immunosuppressed rats
In general, this is a well-organised manuscript reporting a straightforward but interesting study which is relevant to researchers in the field. I would suggest minor revision and my comments are as below:
Title:
This title is very long and not attractive.
I suggest “Pulp-derived stem cells for pulp-dentin complex regeneration: an animal study”
Abstract:
The first (background sentence) can be deleted.
Please use simple and short sentences. For example, The rats were euthanized after 12 weeks. Qualitative evaluations were performed to study intracanal connective tissue, odontoblast-like cells, intracanal mineralized tissue, and periapical inflammatory infiltrate.
Introduction
Please highlight the research gap and put more justifications on this study.
Materials and Methods
State ethics approval in the beginning of this section.
Results
Your photomicrograph is the Key of your results. Enlarge your photomicrographs which can extend and occupy the whole page in width. Currently they are too small to demonstrate the results in particular Figure 3.
Discussion
The author should understand the first language for many researchers and readers not English. Hence, I recommend to use simple and short sentences.
Conclusion
Please start with “In this animal study,” to make your conclusion.
The author should understand the first language for many researchers and readers not English. Hence, I recommend to use simple and short sentences.
Author Response
Title: Pulp-dentin complex regeneration by cell transplantation technique using stem cells derived of the human deciduous teeth: histological and immunohistochemical study in immunosuppressed rats
In general, this is a well-organised manuscript reporting a straightforward but interesting study which is relevant to researchers in the field. I would suggest minor revision and my comments are as below:
Title:
This title is very long and not attractive.
I suggest “Pulp-derived stem cells for pulp-dentin complex regeneration: an animal study”
R:
Although the title may seem long, we believe it contains important information for readers.
Abstract:
The first (background sentence) can be deleted.
R:
It was done
Please use simple and short sentences. For example, The rats were euthanized after 12 weeks. Qualitative evaluations were performed to study intracanal connective tissue, odontoblast-like cells, intracanal mineralized tissue, and periapical inflammatory infiltrate.
Introduction
Please highlight the research gap and put more justifications on this study.
Materials and Methods
State ethics approval in the beginning of this section.
R:
The approval number was already in the text on line 104
Results
Your photomicrograph is the Key of your results. Enlarge your photomicrographs which can extend and occupy the whole page in width. Currently they are too small to demonstrate the results in particular Figure 3.
R:
The images were adapted according to the recommendations of the "instructions to authors"
Discussion
The author should understand the first language for many researchers and readers not English. Hence, I recommend to use simple and short sentences.
Conclusion
Please start with “In this animal study,” to make your conclusion.
R:
It was done
Comments on the Quality of English Language
The author should understand the first language for many researchers and readers not English. Hence, I recommend to use simple and short sentences.
Reviewer 2 Report (Previous Reviewer 4)
Title:
- Please simplify Abstract: - Introduction - L52: sealer and cement - L52: inert materials? I think "inert or bioactive materials": REF: Kharouf, N.; Sauro, S.; Eid, A.; Zghal, J.; Jmal, H.; Seck, A.; Macaluso, V.; Addiego, F.; Inchingolo, F.; Affolter-Zbaraszczuk, C.; Meyer, F.; Haikel, Y.; Mancino, D. Physicochemical and Mechanical Properties of Premixed Calcium Silicate and Resin Sealers. J. Funct. Biomater. 2023, 14, 9. - I didn't see the originality of the present study, please clarify because several studies already investigated this point. Methods: - L132: how radiographically? - Any reference for the final irrigation procedure? - Any statistical analysis? Results: - did the authors investigate all the samples? any change or difference between the samples? - As the authors investigated several samples, thus, please clarify if the results were similaire through the samples Discussion: - Please clarify what is the new of this study compared to previous similar studies - Please more details about the limitations and perspectives References: - goodAuthor Response
Revisor 2
Please simplify Abstract: -
Introduction –
L52: sealer and cement –
L52: inert materials? I think "inert or bioactive materials": REF: Kharouf, N.; Sauro, S.; Eid, A.; Zghal, J.; Jmal, H.; Seck, A.; Macaluso, V.; Addiego, F.; Inchingolo, F.; Affolter-Zbaraszczuk, C.; Meyer, F.; Haikel, Y.; Mancino, D. Physicochemical and Mechanical Properties of Premixed Calcium Silicate and Resin Sealers. J. Funct. Biomater. 2023, 14, 9.
R:
It was done
- I didn't see the originality of the present study, please clarify because several studies already investigated this point.
R:
The originality of these studies lies in the fact that we used human cells in an immunosuppressed xenogeneic model. In this way, the results obtained can support future studies in humans.
Methods: - L132: how radiographically? - Any reference for the final irrigation procedure? - Any statistical analysis?
R:
This study aimed to lead to qualitative and not quantitative results
Results: - did the authors investigate all the samples? any change or difference between the samples? - As the authors investigated several samples, thus, please clarify if the results were similaire through the samples
R:
The results obtained in the two groups studied were very similar, however it was possible to detect some differences in relation to the behavior of the tissues involved
Discussion: - Please clarify what is the new of this study compared to previous similar studies - Please more details about the limitations and perspectives
R:
Discussion of study limitations can be read from line 320 to line 325
References: - good
Reviewer 3 Report (Previous Reviewer 3)
-The title of the study can be modified mentioning the study design as well.
-Please clarify the use of immunosuppressed rats in this study.
-Please expand the future research directions and limitations at the end of the discussion.
-Please mention the source of figures used in this manuscript.
-Please carefully check the use of abbreviations throughout the manuscript, abstract, figures and tables.
minor edits needed
Author Response
Revisor 3
-The title of the study can be modified mentioning the study design as well.
R:
In the title we use in the last sentence: “histological and immunohistochemical study in immunosuppressed rats” which defines the type of study used
-Please clarify the use of immunosuppressed rats in this study.
R:
The protocol published by Lekhooa et al was used, which can be found in position 29 of the paper references
-Please expand the future research directions and limitations at the end of the discussion.
R:
This discussion was held between lines 320 and 325 of the text
-Please mention the source of figures used in this manuscript.
R: Own authorship
-Please carefully check the use of abbreviations throughout the manuscript, abstract, figures and tables.
R: It was done
Reviewer 4 Report (Previous Reviewer 2)
This study performed a histological and immunohistochemical examination of the regeneration of the pulp-dentin complex promoted by transplanting stem cells from the dental pulp of human deciduous teeth into molars of 12 immunosuppressed adult rats with fully formed apices. The manuscript contains five keywords, three figures (with sixteen images), and fifty-seven references. Overall, it is a correct, complete, and well-conducted paper.
The authors have clearly improved this revised version of their manuscript. All suggestions made by the reviewer have been duly addressed. At this moment, the manuscript is acceptable for publication in the journal.
Author Response
Revisor 4
This study performed a histological and immunohistochemical examination of the regeneration of the pulp-dentin complex promoted by transplanting stem cells from the dental pulp of human deciduous teeth into molars of 12 immunosuppressed adult rats with fully formed apices. The manuscript contains five keywords, three figures (with sixteen images), and fifty-seven references. Overall, it is a correct, complete, and well-conducted paper.
The authors have clearly improved this revised version of their manuscript. All suggestions made by the reviewer have been duly addressed. At this moment, the manuscript is acceptable for publication in the journal
R: Thank you very much
Round 2
Reviewer 2 Report (Previous Reviewer 4)
The authors replied to all the comments
This manuscript is a resubmission of an earlier submission. The following is a list of the peer review reports and author responses from that submission.
Round 1
Reviewer 1 Report
Please check my comments below for this manuscript
- The abstract should be written more scientifically. The aim of the study, its results, and its conclusion should have consistency.
- Wordy and passive sentences, and inappropriate word selection make this manuscript difficult to understand.
- The sample size is small for me.
- Simulated body fluid should be more appropriate than the use of PBS for this study.
- Why did the author choose these experimental protocols? Please explain with reference.
- Histological figures are not clearly identical as discussed.
- The discussion is too long. Obtained results are not reflected in the discussion properly.
- Conclusion is exaggerated
Reviewer 2 Report
This study performed a histological and immunohistochemical examination of the regeneration of the pulp-dentin complex promoted by transplanting stem cells from the dental pulp of human deciduous teeth into molars of 12 immunosuppressed adult rats with fully formed apices. The manuscript contains five keywords, three figures (with sixteen images), and fifty-seven references. Overall, it is a correct, complete, and well-conducted paper, although some remarks are made on different sections of the manuscript.
Keywords
The manuscript presents five keywords. For keywords, where possible, please use Medical Subject Headings terms (MeSH Terms). Great! All the keywords are MeSH terms.
References
Total number of the manuscript references: 57.
This is a complete and updated section. References should be checked carefully to transcribe them accurately. Nevertheless, the reference format does not match the journal’s reference format (ACS style guide). According to the journal’s guidelines, references should be described as follows, depending on the type of work:
· Journal Articles:
1. Author 1, A.B.; Author 2, C.D. Title of the article. Abbreviated Journal Name Year, Volume, page range.
· Books and Book Chapters:
2. Author 1, A.; Author 2, B. Book Title, 3rd ed.; Publisher: Publisher Location, Country, Year; pp. 154–196.
3. Author 1, A.; Author 2, B. Title of the chapter. In Book Title, 2nd ed.; Editor 1, A., Editor 2, B., Eds.; Publisher: Publisher Location, Country, Year; Volume 3, pp. 154–196.
· Unpublished materials intended for publication:
4. Author 1, A.B.; Author 2, C. Title of Unpublished Work (optional). Correspondence Affiliation, City, State, Country. year, status (manuscript in preparation; to be submitted).
5. Author 1, A.B.; Author 2, C. Title of Unpublished Work. Abbreviated Journal Name year, phrase indicating stage of publication (submitted; accepted; in press).
· Unpublished materials not intended for publication:
6. Author 1, A.B. (Affiliation, City, State, Country); Author 2, C. (Affiliation, City, State, Country). Phase describing the material, year. (phase: Personal communication; Private communication; Unpublished work; etc.)
· Conference Proceedings:
7. Author 1, A.B.; Author 2, C.D.; Author 3, E.F. Title of Presentation. In Title of the Collected Work (if available), Proceedings of the Name of the Conference, Location of Conference, Country, Date of Conference; Editor 1, Editor 2, Eds. (if available); Publisher: City, Country, Year (if available); Abstract Number (optional), Pagination (optional).
· Thesis:
8. Author 1, A.B. Title of Thesis. Level of Thesis, Degree-Granting University, Location of University, Date of Completion.
· Websites:
9. Title of Site. Available online: URL (accessed on Day Month Year).
Unlike published works, websites may change over time or disappear, so we encourage you create an archive of the cited website using a service such as WebCite. Archived websites should be cited using the link provided as follows:
10. Title of Site. URL (archived on Day Month Year).
For further information about the reference format proposed by the journal, please, consult the following link: https://www.mdpi.com/journal/bioengineering/instructions
Figures
Total number of the manuscript figures: 3.
The figures have appropriate figure legends.
Reviewer 3 Report
The authors conducted a well-designed study however requires further changes to improve.
Please add more details in the introduction section to support the background information and rationale of the study.
Please proofread the manuscript to improve the structure and flow of the text. There are many sentences, which are hard to understand.
Discuss various aspects of other types of stem cells in context, for example, why or why not Human Umbilical Cord Mesenchymal Stem Cells? Information can be added from the recent article: Human Umbilical Cord Mesenchymal Stem Cells: Current Literature and Role in Periodontal Regeneration." Cells 11.7 (2022): 1168.
Please confirm all the figures are original and there are no copyright issues.
Most of images have poor resolution, please check
there is no ethical issues.
Please re-check to assure that all the abbreviations are defined in the text.
Reviewer 4 Report
L1: please delete ''Type of the Paper'' and let ''Article'' Introduction: - L33: PBS should be clarified, please dont use an abbreviation before the complete name - As several studies were already performed in this field, what is the originality of this work, please clarify the originality? - add a null hypothesis Methods: - The subtitles in this part should be numbered - More details about the pulp human stem cells Results: - Figure 1: please higher resolution is recommended, i cannot accept this resolution because there are no details in these images - The same comment in Figure 2, you cannot publish images with low resolution ! - Please add small letters for each panel in the figures and add the small letter in the legend to explain it in details - Figure 3 !!!, very low resolution Discussion: - Please add the limitations of the present study - please accept or reject the proposed hypothesis in the introduction References: - please follow MDPI style, the name of the journal and volume in italic, the year in bold.....